# The Neuroprotective Effects of mGlu1 Receptor Antagonists Are Mediated by an Enhancement of GABAergic Synaptic Transmission via a Presynaptic CB1 Receptor Mechanism

**DOI:** 10.3390/cells11193015

**Published:** 2022-09-27

**Authors:** Elisa Landucci, Rolando Berlinguer-Palmini, Gilda Baccini, Francesca Boscia, Elisabetta Gerace, Guido Mannaioni, Domenico E. Pellegrini-Giampietro

**Affiliations:** 1Department of Health Sciences, Section of Clinical Pharmacology and Oncology, University of Florence, 50139 Florence, Italy; 2The BioImaging Unit, Faculty of Medical Sciences, Newcastle University, Newcastle Upon Tyne NE2 4HH, UK; 3Institute of Physiology, Christian-Albrechts-University Kiel, 24118 Kiel, Germany; 4Department of Neuroscience, Division of Pharmacology, University of Naples Federico II, 80131 Naples, Italy; 5Department of Neuroscience, Psychology, Drug Research and Child Health (NeuroFarBa), Section of Pharmacology and Toxicology, University of Florence, 50139 Florence, Italy

**Keywords:** mGluR, (S)-3,5-dihydroxyphenylglycine (DHPG), (S)-(+)-α-amino-4-carboxy-2-methylbenzeneacetic acid (LY367385), 2-methyl-6-(phenylethynyl)pyridine (MPEP), cannabinoids, ischemia, inhibitory post synaptic current (IPSC), hippocampus

## Abstract

In this study, we investigated the cross-talk between mGlu1 and CB1 receptors in modulating GABA hippocampal output in whole-cell voltage clamp recordings in rat hippocampal acute slices, in organotypic hippocampal slices exposed to oxygen and glucose deprivation (OGD) and in gerbils subjected to global ischemia. CB1 receptor expression was studied using immunohistochemistry and the CA1 contents of anandamide (AEA) and 2-arachidonoylglycerol (2-AG) were measured by LC-MS/MS. Our results show that mGlu1 receptor antagonists enhance sIPSCs in CA1 pyramidal cells and the basal and ischemic hippocampal release of GABA in vivo in a manner that is mediated by CB1 receptor activation. In hippocampal slices exposed to OGD and in ischemic gerbils, mGlu1 receptor antagonists protected CA1 pyramidal cells against post-ischemic injury and this effect was reduced by CB1 receptor activation. OGD induced a transient increase in the hippocampal content of AEA and this effect is prevented by mGlu1 receptor antagonist. Finally, OGD induced a late disruption of CB1 receptors in the CA1 region and the effect was prevented when CA1 pyramidal cells were protected by mGlu1 antagonists. Altogether, these results suggest a cooperative interaction between mGlu1 receptors and the endocannabinoid system in the mechanisms that lead to post-ischemic neuronal death.

## 1. Introduction

The metabotropic glutamate (mGlu) receptors of the I group mGlu1 and mGlu5 are both coupled to polyphosphoinositide hydrolysis [1] and the activation of both subtypes has been demonstrated to amplify excitatory synaptic transmission and contribute to the induction of excitotoxic neuronal death [2,3]. However, experiments performed in our laboratory have shown that they play a differential role in models of postischemic neuronal injury [4]. Specifically, a number of mGlu1 receptor antagonists, but not the mGlu5 antagonist 2-methyl-6-(phenylethynyl)pyridine (MPEP), are able to reduce CA1 pyramidal cell loss in experimental models of cerebral ischemia in vitro and in vivo [5,6,7], presumably via the protective release of GABA in hippocampal terminals [6]. In models of ischemic preconditioning, both mGlu1 and mGlu5 receptors appear to mediate neuroprotective mechanisms [8,9], involving, in this case, the activation of CB1 receptors by 2-arachidonoylglicerol (2-AG) and a decreased AMPA receptor function [10]. Finally, we have recently shown that under selected conditions (i.e., receptor activation with relatively low doses of DHPG or the use of positive allosteric modulators, PAMs), the activation of mGlu5 receptors may also lead to neuroprotection in models of ischemia in vitro via the PI3K-Akt pathway and the reduction of GluA2-lacking AMPA receptors [11].

Multiple observations have suggested that mGlu1 receptor antagonists may induce neuroprotection in different neurodegenerative models by enhancing GABAergic neurotransmission [6,12]. Studies supporting this idea have shown that the activation of mGlu1 receptors may depress inhibitory synaptic transmission in the CA1 hippocampal area [13,14,15], while the potentiation of GABAergic neurotransmission is known to protect CA1 pyramidal cells against post-ischemic neuronal death [16,17]. An alternative and attractive view emerges from data of several laboratories showing that some of the effects of group I mGlu receptors in the CNS are indirectly mediated by a signaling mechanism which involves a functional interaction with the endocannabinoid system [18,19,20,21,22]. In more detail, the activation of group I mGlu receptors has been shown to promote endogenous cannabinoids’ production in postsynaptic cells that retrogradely diffuse and suppress the release of GABA by the activation of CB1 cannabinoid receptors located in the presynaptic terminals of interneurons [23,24]. This mechanism could explain the apparently paradoxical finding that group I mGlu receptor agonists depress (rather than stimulate) inhibitory postsynaptic potentials in the CA1 hippocampal region [25] and [15], given that CB1 receptor antagonists are able to prevent this effect [26,27]. Similarly, a group I mGlu receptor-mediated retrograde inhibition of GABA released through endocannabinoids has been described in CA1 for both short- [28] and long-term [29] forms of synaptic plasticity, as well as for the persistent effects of experimental febrile seizures on the enhancement of hippocampal excitability [30]. More recently, mGlu5 PAMs have been shown to enhance LTP by a mechanism in which mGlu5 in CA1 pyramidal cells induces an endocannabinoid release and CB1-dependent disinhibition [31].

Similarly, we have postulated that the neuroprotective properties of mGlu1 antagonists in models of cerebral ischemia recognize an endocannabinoid-mediated mechanism [4,32]. In order to address this question in greater detail, we have performed a series of electrophysiological, morphological and release studies in hippocampal tissue in vitro and in vivo under control and simulated ischemic conditions, with the final goal of elucidating whether the interplay between mGlu1 receptors and the endocannabinoid system may also be responsible for the mechanisms involved in post-ischemic neuronal death.

## 2. Materials and Methods

### 2.1. Materials

(S)-3,5-dihydroxyphenylglycine (DHPG), (S)-(+)-α-amino-a-methylbenzeneacetic acid (LY 367385), MPEP, 3-MATIDA, WIN 55,212-2, CP 55940, AM 251, bicuculline and gabazine were purchased from Tocris (Bristol, UK). QX 314 was from Alomone laboratories (Jerusalem, Israel) and URB 602 was from Cayman Chemical (Ann Arbor, MI, USA). Tissue culture reagents were obtained from Gibco-BRL (San Giuliano Milanese, MI, Italy) and ICN Pharmaceuticals (Opera, MI, Italy). Propidium iodide (PI) was purchased from Molecular Probes (Leiden, the Netherlands). If not otherwise declared, all cell culture media were purchased from Sigma-Aldrich (Darmstadt, Germany).

### 2.2. Animals

Male Sprague-Dawley rat of age P14-P20 (Harlan, Italy), male and female Wistar rat pups of 7 days of age (Harlan, Italy) and male adult Mongolian gerbils (*Meriones unguiculatus*, Morini, Reggio Emilia, Italy) were used. Animals, housed at 23 ± 1 °C under a 12 h light–dark cycle with lights on at 07:00, were fed a standard laboratory diet with ad libitum access to water. All animal manipulations and procedures were carried out according to the European Community guidelines for animal care and were approved by the Committee for Animal Care and Experimental Use of the University of Florence and authorized by the Italian Ministry of Health (Auth: 205/2012-B).

### 2.3. Electrophysiological sIPSC Recordings from Rat Hippocampal Slices

Preparation of acute hippocampal slices was carried out as previously described [33]. Briefly, young rats (Sprague-Dawley, age P14–P20) were deeply anesthetized with isoflurane and, after decapitation, the brains were rapidly removed and submerged in an ice-cold artificial cerebrospinal fluid (ACSF) of the following composition (mM): 130 NaCl, 24 NaHCO_3_, 3.5 KCl, 1.25 NaH_2_PO_4_, 1 CaCl_2_, 3 MgSO_4_ and 10 glucose saturated with 95% O_2_/5% CO_2_, at pH 7.4. Acute hippocampal slices (300 µm) were obtained by a vibroslicer (Vibratome 1000s, Leica Biosystems, Deer Park, IL, USA) and stored in an incubation chamber at room temperature for about 1 h before use. Slices were individually transferred to the recording chamber of the patch clamp set up and continuously perfused at room temperature (23–26 °C) with carbo-oxygenated aCSF solution, composed of (in mM): NaCl (130), KCl (3.5), NaH_2_PO_4_ (1.25), NaHCO_3_ (25), glucose (10), CaCl_2_ (2) and MgSO_4_ (1). Whole-cell patch recordings were obtained from CA1 pyramidal neurons in voltage clamp and in current clamp configuration using an Axopatch 200A (Axon Instruments) and a pipette of 5–7 MΩ of resistance. The slices were visualized with a microscope (Nikon Eclipse E600FN) equipped for infrared videomicroscopy.

The standard recording solution was composed of (mM): 130 NaCl, 24 NaHCO_3_, 3.5 KCl, 1.25 NaH_2_PO_4_, 1.5 CaCl_2_, 1.5 MgSO_4_ and 10 glucose saturated with 95% O_2_/5% CO_2_, at pH 7.4. All neurons included in this study had a resting membrane potential below –55 mV and an access resistance in the range of 10–20 MΩ that showed only minimal variations during the recordings included in this study. Records were filtered at 5 kHz and digitized at 20 KHz using a Digidata 1322A A/D board. All data were acquired, stored and analyzed on a PC using the pCLAMP, Origin and Graphpad Prism software (Axon Instruments, Foster City, CA, USA, and Microcal Software, Northampton, MA, USA, respectively). The Mini Analysis Program (Synaptosoft Inc., www.synaptosoft.com accessed on 1 July 2010, Leonia, NJ, USA). In all of the experiments, drugs were administered by addition to the superfusing medium and were applied for a sufficient period to allow their full equilibration.

For recordings of spontaneous IPSCs (sIPSCs), electrodes were filled with a KCl-based solution with the following composition (in mM) 140 KCl, MgCl_2_ (1.6), MgATP (2.5), NaGTP (0.5), EGTA (2), HEPES (10) and 5 QX 314, pH = 7.3–7.4. Spontaneous IPSCs were automatically detected. Both the frequency and the peak amplitude of detected events were analyzed. The GABA_A_ receptor blockers bicuculline or gabazine (10 µM) were routinely added at the end of experiments, to verify that the spontaneous IPSCs were completely abolished, confirming that they were GABA_A_-receptor mediated.

### 2.4. Oxygen–Glucose Deprivation (OGD) in Organotypic Hippocampal Cultures

Organotypic hippocampal slice cultures were prepared as previously described [34]. Briefly, the hippocampi were removed from the brains of 7–8 day old Wistar rats and transverse slices (420 µm) were prepared using a McIlwain tissue chopper in a sterile environment. Isolated slices were first placed in ice-cold Hanks’ balanced salt solution (HBSS), supplemented with 5 mg/mL glucose and 1.5% Fungizone^®^ (GIBCO-BRL), then transferred to humidified semiporous membranes (30 mm Millicell-CM 0.4 µm tissue culture plate inserts, Millipore, Italy; 4 per membrane). These were placed in six-well tissue culture plates containing 1.2 mL culture medium containing 50% Eagle’s MEM, 25% heat-inactivated horse serum, 25% HBSS, 5 mg/mL glucose, 1 mM glutamine and 1.5% Fungizone^®^. Slices were maintained at 37 °C, with 100% humidity and 95% air: 5% CO_2_ atmosphere and the medium was changed every three days. Experiments were carried out after 14 DIV. OGD was induced as previously described [35]. Briefly, the slices were exposed to a serum-free medium saturated with 95% N_2_/5% CO_2_ at 37 °C in a gassed incubator equipped with an oxygen controller (BioSpherix, New York, NY, USA). After 30 min, the cultures were transferred to oxygenated serum-free medium containing 5 mg/mL glucose and returned to the incubator under normoxic conditions. Neuronal injury was evaluated 24 h later. Maximal damage was achieved in this system by exposing the slices to 10 mM glutamate for 24 h. Cell injury was assessed using the fluorescent dye propidium iodide (PI), a highly polar compound that is normally excluded from cells with an intact membrane. When the membrane is damaged, PI can enter the cells and upon binding to exposed DNA becomes highly fluorescent. PI (5 µg/mL) was added to the medium at the end of the 24 h post-OGD recovery period. Thirty minutes later, fluorescence was viewed using an inverted fluorescence microscope (Olympus IX-50; Solent Scientific, Segensworth, UK) equipped with a xenon-arc lamp, a low-power objective (4×) and a rhodamine filter. Images were digitized using a video image obtained with a CCD camera (Diagnostic Instruments Inc., Sterling Heights, MI, USA) controlled by software (InCyt Im1TM; Intracellular Imaging Inc., Cincinnati, OH, USA) and subsequently analyzed using the Image-Pro Plus morphometric analysis software (Media Cybernetics, Silver Spring, MD, USA). In order to quantify cell death, the CA1 hippocampal subfield was identified and encompassed in a frame using the drawing function in the image software (ImageJ; NIH, Bethesda, MD, USA) and the optical density of PI fluorescence was recorded. There was a linear correlation between CA1 PI fluorescence and the number of injured CA1 pyramidal cells.

### 2.5. Immunohistochemistry

Immunolabeling in organotypic hippocampal slice cultures was performed as previously described by Boscia et al. [36]. Briefly, organotypic hippocampal slices were washed in 0.1 M phosphate buffer at pH 7.4 and fixed in 4% *w*/*v* paraformaldehyde in 0.1 M PB for 1 h at room temperature. Slices were washed three times in 50 mM Tris- buffered saline pH 7.4 for 30 min between each step. Slices were first blocked in 3% (*w*/*v*) BSA and Triton X-100 0.25% and then incubated with the primary antisera against CB1 receptor (L15, C-terminal, rabbit anti-CB1, polyclonal, 1:4000, generously provided by Dr. Ken Mackie, University of Washington, Seattle, DC, USA). After 48 h, the slices were first incubated in biotinylated horse anti-mouse or goat anti-rabbit IgG (each at 1:200 dilution, Vector Laboratories, Burlingame, CA, USA) for 2 h and then in avidin–biotinylated horseradish peroxidase complex (Elite ABC, 1:300 dilution, Vector) for 1.5 h, always at RT. The peroxidase reaction was developed using 3,3- diaminobenzidine 4-HCl or Ni^2+^-intensified diaminobenzidine as a chromogen and 0.05% H_2_O_2_. After the final wash, sections were dehydrated, coverslipped and processed for microscope analysis. Images were acquired by a digital camera (Coolsnap, Media Cybernetics, Silver Springs, MD, USA) mounted on a Nikon Eclipse 400 microscope. Damage in CA1 following OGD exposure was assessed by bright-field light microscopy.

### 2.6. HPLC-Mass Spectrometer Analysis of Anandamide (AEA) and 2-Arachidonoylglicerol (2-AG) in Organotypic Hippocampal Slices

Dorsal hemisections of organotypic hippocampal slices containing the CA1 region (8 slices per sample) were sonicated in 100 µL CH3CN containing 3.5 pg meta-AEA and 2.5 ng D8-2-AG. After brief centrifugation (13,000× *g* × 10 min at 4 °C), supernatants were diluted in an equal volume of water with 0.1% HCOOH and a 5 µL aliquot was injected into a LC Packings Ultimate capillary HPLC equipped with a CAP-300 calibrator cartridge (Dionex, Sunnyvale, CA, USA) and coupled to an API 365 PE-Sciex triple quadrupole (QqQ system) mass spectrometer equipped with a TurboIonspray interface from Applied Biosystems (Monza, Italy). The HPLC separation was performed utilizing a 50 × 0.3 mm, 3 µm Luna C18 column (Dionex) at a flow rate of 5 µL/min with a programmed gradient elution using two solvents containing varying concentrations of H_2_O, CH_3_CN and 0.1% HCOOH. Fragmentation was accomplished with a collision energy of 27 eV, N_2_ was used as the collision gas at a collision gas thickness (CGT) of 2.6 × 10^11^ molecules cm^−2^. The mass spectrometry analysis was performed in positive ion mode with MRM technique. The monitored ion transitions were m/z 348.3 → 62.0 for AEA, m/z 362.2 → 76.0 for meta-AEA, m/z 379.4 → 287.0 for 2-AG and m/z 387.4 → 294.0 for D_8_-2-AG. These transitions were selected on the basis of triple quadrupole collision-induced dissociation product ion spectra of the analytes. The dwell time for each transition was 250 ms, resulting in a sampling of more than 15 points per chromatographic peak. The analyte abundance was evaluated by measuring the chromatographic peak height of selected product ions by means of MaxQuant software, version 1.2, (Max Planck Institute of Biochemistry, www.maxquant.org accessed in 1 July 2010, Martinsried, Germany).

### 2.7. Microdialysis Studies in Freely Moving Gerbils and HPLC Quantitation of Glutamate and GABA

Male adult Mongolian gerbils (*Meriones unguiculatus*, Morini, Reggio Emilia, Italy) weighing 60–80 g were anesthetized with chloral hydrate (300 mg/kg, i.p.) and positioned in a stereotaxic frame for the implantation of a transcerebral microdialysis tube (internal diameter: 220 μm; external diameter: 310 μm) in their dorsal hippocampi (coordinates for fiber in- and outlet: anterior–posterior from bregma: −1.8 mm; from skull surface: −1.5 mm), as previously described [37]. After at least 24 h recovery, the membranes were perfused with an iso-osmotic solution (NaCl, 155 mM; KCl 5.5 mM; CaCl_2_ 2.3 mM, pH = 7.2) at a flow rate of 3.5 μL/min. After a washout period of approximately 2 h, 4 fractions were collected every 15 min to determine the basal output. The recovery of GABA in the probes was approximately 45%. The membranes were then perfused for 1 h with a solution containing 1 mM LY 367385 alone or with 50 µM WIN 55,212-2 and then again with the control solution in order to monitor the recovery of the basal output. The concentrations of the drugs were selected assuming that only 10–20% of these diffuse from the microdialysis probe into the surrounding tissue. At the end of the experiments, a solution containing 100 mM KCl was injected through the dialysis fiber to assess the functional integrity of the preparation: only if the solution increased the basal output of GABA by at least 2-fold would the experiment was accepted for analysis [38].

Samples were kept on ice during the collection period and stored at −20 °C until assayed. GABA was measured by HPLC separation and fluorometric detection after precolumn derivatization with *o*-phthaldialdehyde and 2-mercapto-ethanol. The procedure was essentially the same as described in Pellegrini-Giampietro et al. [39].

### 2.8. Microdialysis Experiments in Gerbils Subjected to Transient Global Ischemia and Assessment of CA1 Pyramidal Cell Injury

Twenty four hours after the implantation of transcerebral microdialysis tubes, animals were subjected to transient global ischemia, as previously described [40]. Briefly: gerbils were anesthetized with a mixture of 2% isoflurane, 75% nitrogen and 20% oxygen. Through a ventral midline neck incision, both common carotid arteries were isolated and occluded for 5 min using micro-arterial clamps. At the end of the occlusion period, the clamps were released allowing the restoration of carotid blood flow and the incision was sutured. Body temperature was monitored and maintained at 37 °C with a rectal thermistor and heating pad until the animals had fully recovered from anesthesia. In this set of experiments, the collection of microdialysis fractions started 60 min before the occlusion and was protracted up to 100 min after circulation was restored. Hence, fractions were collected every 15 min, except for the fraction immediately after the occlusion, that was collected every 5 min.

Seven days after the ischemic insult, the gerbils were sacrificed by decapitation, their brains rapidly removed and frozen in dry ice. Coronal sections (20 µm) were cut in a cryostat and stained with toluidine blue. At least four microscopic sections from hippocampal areas between 1 and 3 mm anterior or posterior to the location of the microdialysis tube were analyzed for each animal.

### 2.9. Statistical Analysis

All numerical data are expressed as mean ± SEM. Data were analyzed statistically by paired or unpaired Student’s *t test*, the Kolmogorov–Smirnov test or ANOVA followed by Tukey’s w test. A value of *p* < 0.05 was considered statistically significant. All statistical calculations were performed using Prism 5 for Windows (Graph Pad Software, San Diego, CA, USA).

## 3. Results

### 3.1. mGlu1 but Not mGlu5 Competitive Antagonists Increase sIPSCs in CA1 Pyramidal Cells

Inhibitory neurotransmission was studied in spontaneously occurring pharmacologically isolated inhibitory post-synaptic currents (sIPSCs) in CA1 pyramidal cells. Action potential-dependent sIPSCs were measured in pyramidal neurons under voltage clamp at −70 mV with a chloride-based internal solution containing QX 314 in order to avoid the neuronal generation of action potentials (see Methods).

Under our experimental conditions, the application of the competitive mGlu1 receptor antagonist LY367385 (100 and 300 µM) dose-dependently increased the sIPSC frequency (Figure 1A–C left panel and 1D upper panels) and amplitude (Figure 1A,C right panel and 1D upper panels). These effects were reversible following drug wash-out and were also obtained using 3-MATIDA (300 µM), another selective mGlu1 receptor competitive antagonist [7] (Figure 1D, bottom panels). The kinetics of the sIPSCs did not change after the application of LY367385 (Figure 1A inset): the decay time constant was analyzed in a subset of cells showing no significant difference between control cells and following LY367385 application (54.4 ± 2.6 ms versus 57.8 ± 3.3, *n* = 4 in control and LY367385, respectively, Student’s paired *t test*). LY367385 produced a leftward shift of the cumulative inter-event interval distribution and a shift in sIPSC amplitude toward a bigger cumulative amplitude (*p* < 0.001; Kolmogorov–Smirnov test) indicating an increase in both the frequency and amplitude of sIPSCs (Figure 1C). Interestingly, under our experimental conditions, MPEP (10 µM), a selective noncompetitive mGlu5 receptor antagonist, did not modify neither the frequency nor the amplitude of sIPSCs (Figure 1D, bottom panels), suggesting that only the blockade of mGlu1 receptors was able to produce a modulation of sIPSCs. The increase in frequency and amplitude of GABA_A_-mediated sIPSCs induced by LY367385 was not present when tetrodotoxin, a selective blocker of voltage-activated Na^+^ channels, was added to the medium (1.27 ± 0.37 Hz vs. 1.25 ± 0.4 Hz and 63 ± 3 pA vs. 60 ± 1.8 pA in controls and LY367385, respectively; *n* = 4; *p* > 0.05, Student’s *t test*). Moreover, LY367385 effects were specific for CA1 pyramidal cells since stratum oriens-alveus interneurons did not show any changes in both the frequency and amplitude of sIPSCs (0.98 ± 0.52 Hz vs. 0.96 ± 0.54 Hz; 35.25 ± 4.82 pA versus 41.98 ± 4.94 pA in controls and LY367385, respectively; *n* = 5; *p* > 0.05, Student’s *t test*) and no depolarizing effect was present following LY367385 application. Finally, the effects of LY367385 on sIPSCs did not desensitize over time and were fully reversible upon drug wash-out.

### 3.2. The Increase in sIPSCs Induced by mGlu1 Receptor Antagonists Is Prevented by CB1 Receptor Activation

A functional interaction exists between group I mGlu receptors and the endocannabinoid system in the modulation of inhibitory synaptic transmission. In particular, it has been repeatedly demonstrated that group I mGlu-induced IPSCs suppression is mediated by endocannabinoids [28,41]. Therefore, we reasoned that the increase in sIPSCs frequency and amplitude produced by the mGlu1 antagonist LY367385 in our system could also be dependent on an endocannabinoid mechanism. First, we tested whether the application of CB1 receptor agonists could modulate sIPSCs in the CA1 hippocampal area: in accord with previous observations [42,43], the application of the selective CB1 receptor agonist WIN55,212-2 at 30 µM was able to decrease the frequency of sIPSCs (4.57 ± 1.45 Hz in control versus 3.67 ± 1.24 Hz with WIN55,212-2, *n* = 9; *p* < 0.05 versus control), suggesting a presynaptic modulation of GABA release via CB1 receptor activation. Then, we observed that the increasing effects of LY367385 on sIPSC frequency and amplitude were reverted following the application of the two CB1 receptor selective agonists CP55,490 at 10 µM (Figure 2A–C) and WIN 55,212-2 at 30 µM (Figure 2D). Finally, the effect of LY367385 on the sIPSC frequency was blocked by the selective inhibitor of 2-AG’s degrading enzyme monoacylglycerol lipase URB602 (100 µM) (Figure 2E,F), suggesting a contribution of 2-AG in the modulatory effect of LY367385 on GABAergic neurotransmission.

### 3.3. The Neuroprotective Effects of mGlu1 Receptor Antagonists against OGD Toxicity in Organotypic Hippocampal Slices Are Prevented by CB1 Receptor Activation

We and other laboratories have repeatedly demonstrated that mGlu1 receptor antagonists such as LY367385 and 3-MATIDA are able to attenuate post-ischemic injury in vitro and in vivo [4]. As the activation of the endocannabinoid system was required for the mGlu1-mediated enhancement of the GABAergic neurotransmission observed in pyramidal cells, we questioned whether cannabinoids could also be involved in determining the neuroprotective effects of mGlu1 antagonists in organotypic hippocampal slices exposed to 30 min OGD. As previously reported [7], LY367385 (300 µM) significantly reduced CA1 injury when present in the incubation medium during OGD and the subsequent 24 h recovery period (Figure 3A,C). This neuroprotective effect was reverted by the CB1 receptor agonist WIN 55,212-2 at 30 µM, but not by the CB1 antagonist AM 251 at 1 µM (Figure 3A,C), suggesting a role for CB1 receptors in modulating the neuroprotective effects of mGlu1 antagonists. When organotypic hippocampal slices were exposed to a shorter period of OGD (20 min), the selective mGlu1/5 agonist DHPG at 100 µM was able to significantly exacerbate CA1 injury (Figure 3B,D). In this case, the CB1 antagonist AM 251 but not the CB1 agonist WIN 55,212-2 was able to attenuate DHPG-induced neurotoxicity (Figure 3B,D), confirming an interaction between the functional actions of mGlu1 and CB1 receptors.

### 3.4. The Disruption of Cb1-Like Immunoreactivity in Organotypic Hippocampal Slices Exposed to OGD Is Reduced by mGlu1 Receptor Antagonists

In order to investigate the expression of the CB1 receptor in organotypic hippocampal slices exposed to OGD, we examined the pattern of immunoreactivity observed with an anti-CB1 antibody. As previously reported [36], we observed an extensive perisomatic axonal meshwork in the stratum pyramidale, with a dense plexus of CB1-immunoreactive fibers that surrounded the immunonegative CA1 pyramidal cells (Figure 4B–D). As observed 24 h after reoxygenation, OGD induced an evident disruption of CB1 receptor immunoreactivity in the CA1 region (Figure 4F–H). Nonetheless, a CB1 immunosignal was clearly detected in the isolated CB1-positive interneurons which appeared intensely immunostained throughout the stratum radiatum of the CA1 subfield (Figure 4G). When slices were exposed to OGD in the presence of the mGlu1 antagonist 3-MATIDA, the reduction of CB1 immunoreactivity observed after OGD was partially prevented (Figure 4J–L).

### 3.5. The mGlu1 Antagonist LY367385 Prevents the Formation of Endocannabinoids in the CA1 Region of Organotypic Hippocampal Slices Exposed to OGD

We have reported that CB1 receptor antagonists are neuroprotective in organotypic hippocampal slices exposed to OGD, whereas CB1 agonists exacerbate OGD injury [44] in a similar, but not additive manner to what observed with DHPG, suggesting that CB1 receptor activation may also contribute to the neurodegenerative mechanisms following cerebral ischemia. Therefore, we measured the levels of the endocannabinoids anandamide (AEA) and 2-arachidonoylglycerol (2-AG) produced in dorsal hemisections (containing the CA1 area) of organotypic hippocampal slices following OGD. Under basal conditions, the content of AEA in slices was 13.5 ± 12 pg/mg of protein and that of 2-AG was 341 ± 69 ng/mg of protein. Immediately after OGD, the formation of AEA increased four-fold, but 3 and 24 h later, the levels returned to baseline (Figure 5A). On the other hand, the formation of 2-AG was not modified by OGD (Figure 5B). Interestingly, both the immediate and transient increase in AEA following OGD and the content of 2-AG were reduced by incubation with the mGlu1 antagonist LY367385 (Figure 5A,B), suggesting that the production of endocannabinoids during OGD requires the activation of the mGlu1 receptors.

### 3.6. The Hippocampal Output of GABA and the Neuroprotective Effects Induced by mGlu1 Receptor Antagonists in Ischemic Gerbils Are Prevented by CB1 Receptor Activation

We have reported that mGlu1 receptor antagonists are neuroprotective and enhance the concentration of GABA in the hippocampal dialysate of gerbils subjected to global ischemia [5,6]. In order to establish whether endocannabinoids could also be involved in this mechanism in vivo, we used transverse microdialysis to study the GABA output from the hippocampus of gerbils under control and global ischemic conditions. After 2 h of equilibration, the basal output of GABA from the microdialysis probe was measured and remained constant for approximately 1 h. Application through the microdialysis perfusion fluid of the mGlu1 receptor antagonist LY367685 (1 mM) induced an up to two-fold increase in the mean basal output of GABA (Figure 6A). When the CB1 receptor agonist WIN 55,212-2 (50 µM) was perfused along with LY367385, the GABA output remained at the basal levels. Transdialytic perfusion with LY367385 (1 mM) during and following 5 min of bilateral carotid occlusion produced a further increase in the output of GABA that was maximal 15 min after circulation was restored (21.0 ± 2.0-fold over mean basal levels, Figure 6B). Interestingly, when WIN 55,212-2 (50 µM) was transdialytically co-perfused with LY367385, the LY367385-induced ischemic increase of the GABA output was completely abolished, suggesting a crucial role for CB1 receptor activation in the modulation of GABA release by LY367385 both under normal and ischemic conditions in vivo.

Moreover, histological analysis of the same animals confirmed the attenuation of CA1 pyramidal post-ischemic cell damage previously observed with mGlu1 receptor antagonists in vivo [5,6]. Transdialytic perfusion of ischemic gerbils with 1 mM LY 367385 produced a dramatic reduction of post-ischemic CA1 pyramidal cell loss (from 95 ± 1 to 7 ± 3%) while co-perfusion with 50 µM WIN 55,212-2 abolished the neuroprotective effect of LY367835 (Figure 6C–J).

## 4. Discussion

Our results show that the neuroprotective effects of mGlu1 receptor antagonists in models of ischemia in vitro and in vivo are mediated by an enhancement of GABAergic neurotransmission that appears to be regulated by the endocannabinoid activation of presynaptic CB1 receptors. We have demonstrated multiple interactions between mGlu1 receptors and the endocannabinoid system by showing that: (i) mGlu1 receptor antagonists enhance sIPSCs in CA1 pyramidal cells and the basal and ischemic hippocampal release of GABA in vivo and these effects are reduced by CB1 receptor activation; (ii) mGlu1 receptor antagonists protect CA1 pyramidal cells against post-ischemic injury in vitro and in vivo and this effect is reduced by CB1 receptor activation; (iii) OGD induces a transient increase in the hippocampal content of AEA and this effect is prevented by mGlu1 receptor antagonist; (iv) OGD induces a late disruption of CB1 receptors in the CA1 region and the effect is prevented when CA1 pyramidal cells are protected by mGlu1 antagonists.

mGlu1 receptor antagonists have repeatedly been shown to attenuate post-ischemic injury by enhancing GABA-mediated neurotransmission. For example, they increase the concentrations of GABA in the hippocampal dialysate of gerbils subjected to global ischemia [5] and, in hippocampal slices exposed to OGD, 3-MATIDA, as well as GABA_A_ and GABA_B_ receptor agonists, reduce CA1 injury while GABA receptor antagonists partially prevent this effect [6]. Similarly, LY367385 and CPCCOEt increase GABA extracellular levels of freely moving rats in the corpus striatum in parallel with a reduction in NMDA neurotoxicity [12]. Furthermore, the neuroprotective effects of mGlu1 receptor antagonists are occluded by the previous application of GABA and SKF89976A (a GABA transporter inhibitor) and are prevented by the GABA_A_ and GABA_B_ receptor antagonists in cultured neuronal cells exposed to NMDA [12]. Finally, the reduction of spontaneous epileptiform activity induced by 3-MATIDA was similarly reproduced by a GABA receptor agonist and reverted by an antagonist in mouse cortical wedges [6]. All together, these data highlight a common GABA-mediated mechanism, which involves the release of GABA and the stimulation of GABA receptors, for the neuroprotective effects of mGlu1 receptor antagonists.

These findings suggest a possible presynaptic localization for mGlu1 receptors in GABAergic terminals inhibiting the release of GABA. Functional data support the existence of presynaptic mGlu1 receptors modulating neurotransmitter release in the hippocampus [45] and neocortex [46]. Moreover, the stimulation of group I mGlu receptors in the hippocampal CA1 and other brain areas increases principal cell excitability possibly through a presynaptic inhibition of GABA release from interneurons [13,15,47,48]. The presynaptic inhibition of neurotransmitter release by mGlu1 receptors could be mediated by a suppression of Ca^2+^ currents via N- or P/Q-type channels [49] or by the activation of a Ca^2+^-dependent K^+^ conductance [50]. To date, only group I receptors of the mGlu5 subtype have been detected at a presynaptic level [51], but two reports provide electron microscopy evidence for mGlu1a and mGlu5 staining in GABAergic presynaptic terminals and preterminal GABAergic axons in the substantia nigra [48,52]. The activation of these presynaptic receptors appears to be responsible for the decrease in inhibitory transmission observed in this area [48] and a similar mechanism may be operative in the hippocampus. Although anatomical studies have essentially demonstrated that mGlu1α receptors, at least in the hippocampus, are expressed almost exclusively in non-principal cells, functional evidence exists for the presence of mGlu1 together with mGlu5 receptors in CA1 pyramidal cells [15,53]. Moreover, mGlu1β and mGlu1d splice variants have been shown to be expressed in hippocampal principal neurons, albeit not abundantly in CA1 [54,55]. Hence, it is also possible that a presynaptic decrease in inhibitory transmission can be indirectly mediated by mGlu1 receptors located postsynaptically in principal neurons. Our electrophysiological recordings in the presence of LY367385 confirm that a tonic stimulation of mGlu1 receptors exists in interneurons of acute (non-ischemic) rat hippocampal slices [56]; moreover, our data show that the kinetics of sIPSCs do not change following the application of LY367385, suggesting that the increase in the IPSC frequency and amplitude observed after the blockade of the mGlu1 receptors is probably not due to changes in the phosphorylation or subunit composition of GABA_A_ receptors, but rather to an increased release of GABA at the presynaptic level and/or to an increase in the number or in the open probability of channels. Our microdialysis experiments suggest that the former could be a plausible mechanism for the observed effects of LY367385 on IPSCs.

Emerging studies from several laboratories have revealed that some of the effects of group I mGlu receptors in the CNS are indirectly mediated by a novel signaling mechanism which entails a functional interaction with the endocannabinoid system [18,19,20,21,22]. Specifically, the activation of group I mGlu receptors has been shown to promote in postsynaptic cells the production of endogenous cannabinoids that retrogradely diffuse and suppress the release of GABA by the activation of CB1 cannabinoid receptors located on the presynaptic terminals of interneurons [24,27]. This mechanism appears to be responsible for the above-mentioned finding that group I mGlu receptor agonists depress (rather than stimulate) inhibitory postsynaptic potentials in the CA1 hippocampal region, given that CB1 receptor antagonists are able to prevent this effect [26,27]. Similarly, a group I mGlu receptor-mediated retrograde inhibition of GABA release through endocannabinoids has been described in CA1 for both short- [28] and long-term [29] forms of synaptic plasticity, as well as for the persistent effects of experimental febrile seizures on the enhancement of hippocampal excitability [30]. In our study, mGlu1 antagonist significantly prevented the OGD-induced injury observed in the CA1 subregion and this effect was completely reverted by the CB1 receptor agonist WIN 55,212-2, but not the antagonist AM 251, thus suggesting that the neuroprotection afforded by the mGlu1 antagonist is mediated by CB1 receptors.

In the CA1 hippocampal region, both mGlu1α and CB1 receptors are characteristically expressed in GABAergic interneurons. As described, mGlu1α receptors are enriched in interneurons of the stratum oriens–stratum lacunosum moleculare that contain somatostatin, but are also present in interneurons expressing vasoactive intestinal peptide (VIP) and/or calretinin and in a subpopulation of cholecystokinin (CCK)-immunopositive interneurons [57]. On the other hand, CB1 receptors are primarily expressed in CCK-immunoreactive basket cells of the hippocampus [58], but they have also been described to be partially co-localized with neurons containing the calcium binding proteins calretinin and calbindin in the same area [59]. In a previous study [36], we performed a double-labeling confocal fluorescence analysis with specific mGlu1α and CB1 receptor antibodies both in rat organotypic hippocampal slice cultures and in hippocampal sections from adult rat brains. Our results showed that a subset of interneurons, mainly located in the stratum radiatum, was double-labeled for both mGlu1α and CB1 receptors. It has been recently shown that persistently active cannabinoid receptors switch off (mute) the output of a unique class of hippocampal interneurons [60] which could be recruited once the endocannabinoid tone is attenuated by mGlu1 antagonists. In this study, the typical plexus of CB1-immunoreactive fibers that surround the immunonegative CA1 pyramidal cells appeared to be decreased and disrupted in hippocampal slices 24 h after OGG, presumably because of the degeneration of the target CA1 pyramidal cells. Interestingly, the neuroprotection of pyramidal cells with a mGlu1 receptor antagonist restored the immunoreactive decoration in the CA1 subregion.

Finally, we observed a transient increase in the formation of AEA, but not 2-AG, in organotypic hippocampal slices exposed to OGD. This is in accord with previous observations detecting increases in AEA rather than 2-AG in models of cerebral ischemia [61] and with previous results showing that CB1 agonists exacerbate OGD injury whereas CB1 antagonists are neuroprotective in this model [44]. Notably, both the ischemic levels of AEA and the content of 2-AG were reduced by the mGlu1 receptor antagonist LY 367385, suggesting that AEA production during OGD might be regulated by mGlu1 receptors either in neurons or in astrocytes [62].

Altogether, these results point out to a cooperation between mGlu1 receptors and the endocannabinoid system in the mechanisms that lead to GABAergic neurotransmission and post-ischemic neuronal death. It appears as though the protective effects of mGlu1 receptor antagonists are mediated by a mechanism that overcomes the “synaptic circuit break” operated by endocannabinoids on GABAergic transmission [58,63]. We would like to propose a polysynaptic GABAergic disinhibition model (Figure 7) as a possible explanation for our data, in which mGlu1 are postsynaptic and CB1 receptors presynaptic in different interneuron populations connected in series.

The model is based on a number of observations, including: (i) the prominent anatomical localization of mGlu1 receptors in dendrites and somata of somatostatin-positive interneurons of the stratum oriens-lacunosum moleculare [57], (ii) the primary expression of CB1 receptors in the presynaptic terminals of CCK-immunopositive interneurons innervating the perisomatic region of pyramidal cells [58] and in astrocytes [62], (iii) the demonstration that stratum oriens-lacunosum moleculare GABAergic interneurons target and inhibit dendrites of other interneurons such as basket cells in the stratum radiatum [58], which themselves inhibit the proximal dendrites of CA1 pyramidal cells providing the indirect disinhibition of the excitatory inputs from the Schaffer collateral pathway [64], (iv) our experiments showing that tetrodotoxin prevents the increase in synaptic inhibition induced by LY367385, suggesting an indirect effect on presynaptic GABAergic neurons that requires an action potential-driven GABA release, and (v) our finding that mGlu1 antagonists and CB agonists were able to directly modulate the hippocampal release of GABA. However, because we have observed changes in both the frequency and amplitude of sIPSCs with mGlu1 antagonists and CB agonists, the contribution of changes in postsynaptic GABA_A_ receptors located in pyramidal cells cannot be ruled out.

Our hypothesis predicts that during ischemia, the hyperactivation of mGlu1 receptors promoted by the dramatic increase in release the of glutamate will increase the firing of the first interneuron and lead to the inhibition of the second one (possibly a basket cell) innervating pyramidal cells, resulting in a reduction in the release of GABA and degeneration of the CA1 pyramidal cells. The formation of AEA will exacerbate this mechanism by further reducing the release of GABA. mGlu1 antagonists will counteract this mechanism and, therefore, increase the net output of GABA upon pyramidal cells and provide neuroprotection, CB1 agonists will reduce the output from basket cell terminals and prevent this effect. Further studies are required to determine the validity of these hypotheses. The clarification of these mechanisms is expected to provide new insight into the possible targets for new therapeutic interventions for stroke and other ischemia-related syndromes.

## Figures and Tables

**Figure 1 cells-11-03015-f001:**
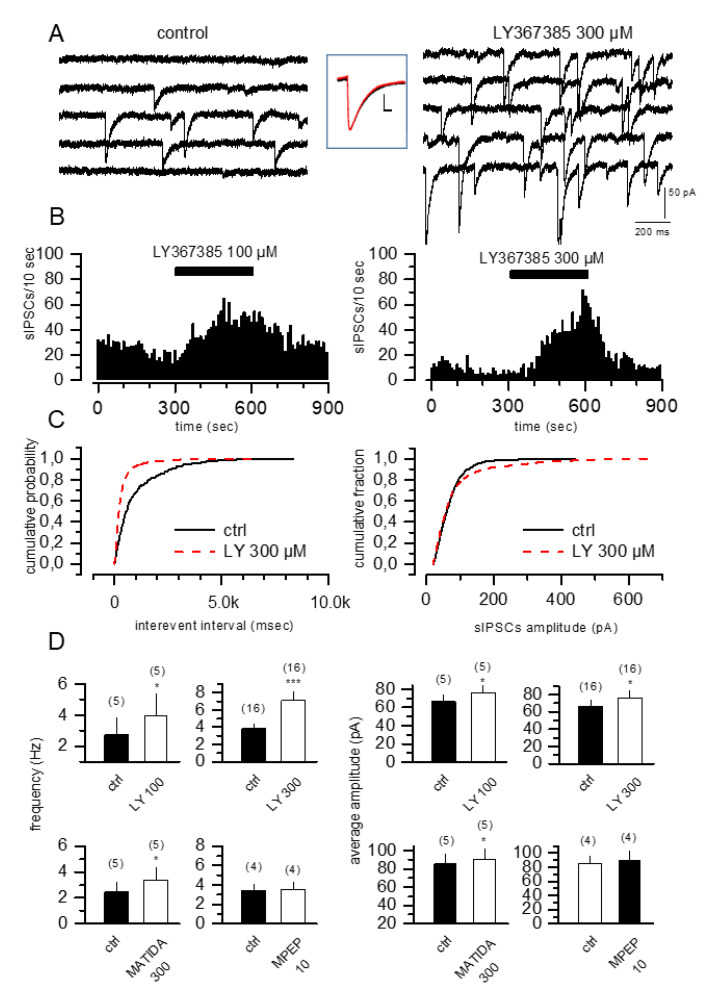
mGlu1 but not mGlu5 receptor antagonists selectively increase the frequency and the amplitude of sIPSCs in CA1 pyramidal cells. (**A**) sIPSCs recorded from CA1 pyramidal single cells before and during LY367385 (300 µM) application. The inset shows superimposed traces of averaged sIPSCs recorded from CA1 pyramidal cell (control, black line: *n* = 420 events; LY367385, red line: *n* = 1400 events). Scale bar in inset 10 ms and 100 pA. (**B**) Frequency time courses recorded from single cells showing the reversible increase in sIPSCs frequency mediated by application of 100 µM (**left**) and 300 µM (**right**) LY367685. (**C**) Cumulative probability plots demonstrating the effect of LY367385 on sIPSCs interevent interval and amplitude [Kolmogorov–Smirnov test; *p* < 0.001]. (**D**) Bar graphs showing the increase in frequency and amplitude of sIPSCs induced by different concentrations of LY367385 (100 and 300 µM), 3-MATIDA (300 µM) and MPEP (10 µM). The number of cells tested is in parenthesis; values are the means ± SEM. * *p* < 0.05, *** *p* < 0.001 versus control, Student’s *t test*.

**Figure 2 cells-11-03015-f002:**
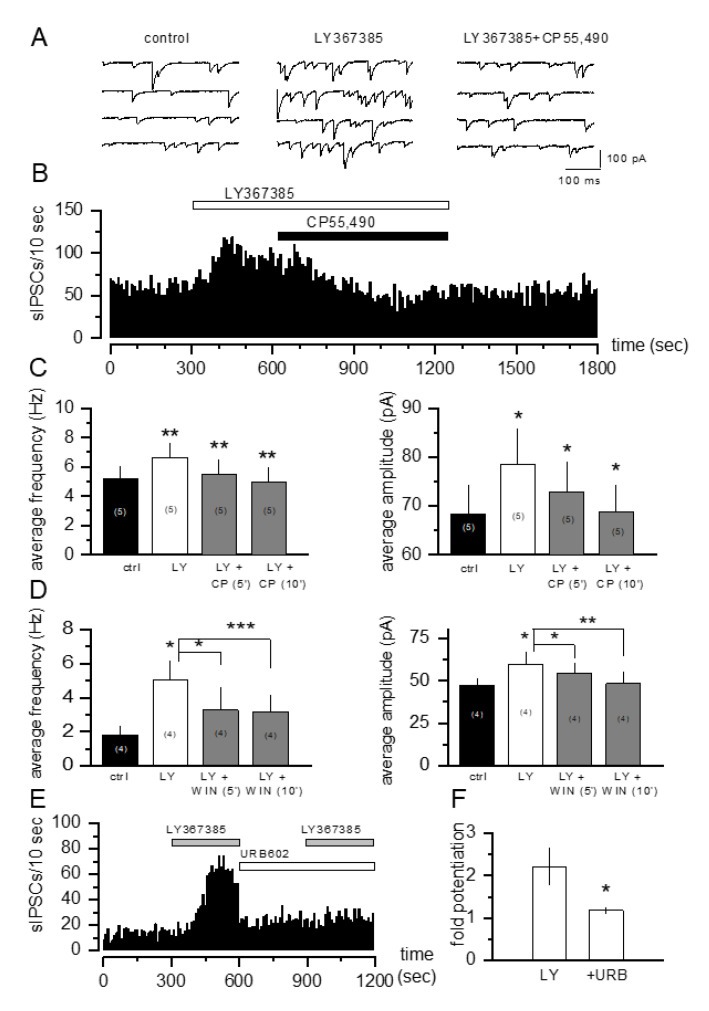
The CB1 receptor agonist CP55,490 and the monoacylglycerol lipase inhibitor URB602 revert the increase in sIPSC frequency and amplitude induced by the mGlu1 receptor antagonist LY367385. (**A**) sIPSCs recorded from CA1 pyramidal single cells under control condition (**left**), during bath application of 300 µM LY367385 (**middle**) and during bath application of LY367385 plus 10 µ CP55,490 (**right**). (**B**) Frequency time course recorded from a single cell showing that 10 µM CP55,490 reverts the increase in sIPSC frequency induced by LY367385. (**C**,**D**) Bar graphs showing that 10 µM CP55,490 and WIN 55,212-2 revert the increase in sIPSC frequency (**left panel**) and amplitude (**right panel**) induced by LY367385, 5 and 10 min after application. (**E**) Frequency time course recorded from a single cell showing that 100 µM URB602 reverts the increase in sIPSC frequency induced by LY367685. URB602 was preincubated for 5 min and no changes of sIPSC frequency were noticed. (**F**) Bar graphs showing the reduction of the effects of LY367385 on sIPSC frequency following URB602 pre-incubation. The number of cells tested is in parenthesis; values are the means ± SEM, *n* = 4. * *p* < 0.05, ** *p*<0.01 and *** *p*<0.001 vs. LY367385 potentiation, Student’s *t test*.

**Figure 3 cells-11-03015-f003:**
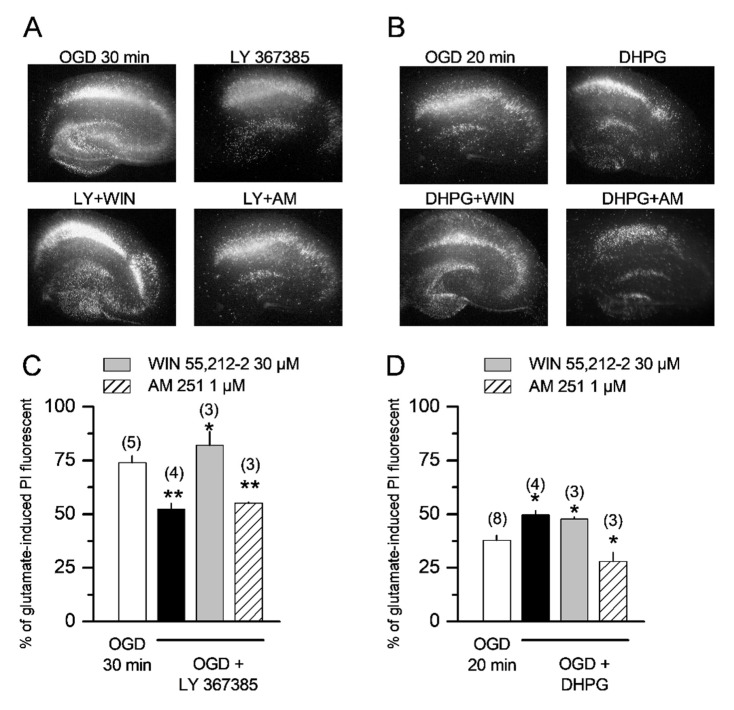
The neuroprotective effects of the mGlu1 antagonist LY367385 and the toxic effects of the mGlu1 agonist DHPG are reverted by CB1 receptor ligands in organotypic hippocampal slices exposed to OGD. (**A**,**C**) Qualitative and quantitative analysis of organotypic hippocampal slice exposed to OGD, displaying a selective increase in PI fluorescence in the CA1 subregion. The CB1 agonist WIN 55,212-2, but not the CB1 antagonist AM 251, reverts the neuroprotective effects of 300 µM LY367385 against 30 min OGD injury. (**B**,**D**) Qualitative and quantitative analysis of organotypic hippocampal slice exposed to OGD, showing that the CB1 antagonist AM 251, but not the CB1 agonist WIN 55,212-2, reverts the exacerbation of 20 min OGD injury induced by 100 µM DHPG. Drugs were present in the incubation medium during OGD and the subsequent 24 h recovery period. Bars represent the mean ± SEM, the number of experiments (run in quadruplicate) is shown in parenthesis. * *p* < 0.05 and ** *p* < 0.01 vs. OGD, respectively (ANOVA + Tukey’s w test).

**Figure 4 cells-11-03015-f004:**
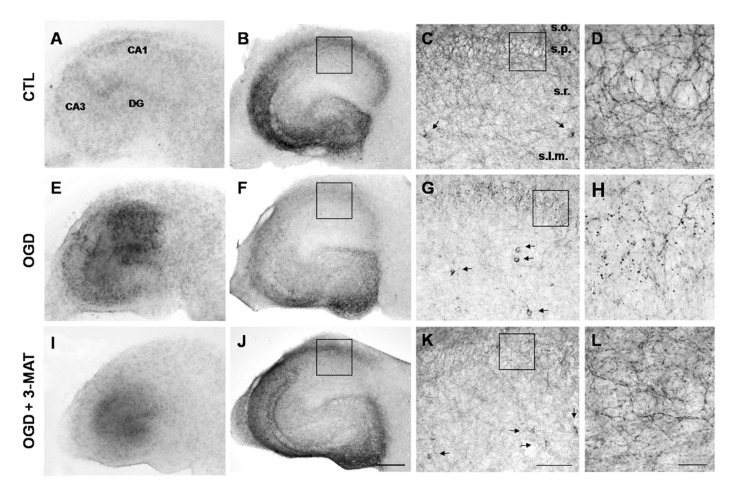
CB1 receptor expression in organotypic hippocampal slices exposed to OGD. (**A**,**E**,**I**) Bright-field transmission images of organotypic hippocampal slices displaying CA1 injury 24 h after 30 min OGD and neuroprotection with the mGlu1 antagonist 3-MATIDA (3-MAT, 100 µM). (**B**,**F**,**J**) CB1-like immunoreactivity in neurons of all hippocampal areas. OGD produces 24 h later a reduction of the dense plexus of immunolabeling mainly in CA1, that is prevented by 3-MATIDA. (**C**,**G**,**K**) Higher magnification of frames depicted in B-F-J displays an intense network of CB1-positive processes in stratum pyramidalis (s.p.) that is severely damaged by OGD and protected by 3-MATIDA. Conversely, isolated stratum radiatum (s.r.) CB1-positive interneurons (arrows) are not affected by OGD. (**D**,**H**,**L**) Higher magnification of frames depicted in (**C**,**G**,**K**) displays dramatic disruption of the CB1-like immunoreactive perisomatic axonal meshwork in stratum pyramidalis that is partially prevented by 3-MATIDA. Scale bars: 400 µm in (**A**,**E**,**I**,**B**,**F**,**J**), 100 µm in (**C**,**G**,**K**) and 20 µm in (**D**,**H**,**L**). Other abbreviations: s.o., stratum oriens; s.l.m., stratum lacunosum molaeculare.

**Figure 5 cells-11-03015-f005:**
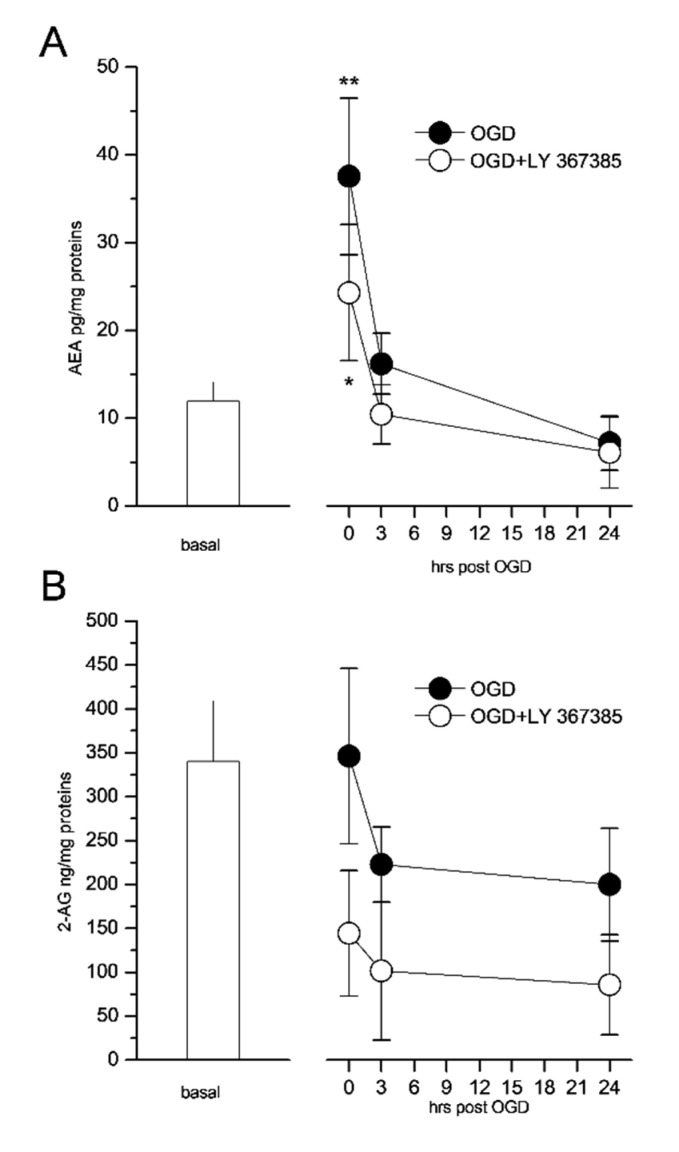
The mGlu1 receptor antagonist LY367385 reduces the formation of the endocannabinoids AEA and 2-AG in organotypic hippocampal slices exposed to 30 min OGD. (**A**) The contents of AEA were transiently increased immediately after OGD and the returned to basal levels. LY367385 was able to reduce the transient increase. (**B**) The contents of 2-AG were not changed after OGD, but LY367385 was still able to reduce its levels. AEA and 2-AG were measured in 8 dorsal hemisections/sample. LY367385 was present in the incubation medium during the 30 min OGD exposure and the subsequent recovery period. * *p* < 0.05 and ** *p* < 0.01 vs. OGD alone and control, respectively (ANOVA + Tukey’s w test).

**Figure 6 cells-11-03015-f006:**
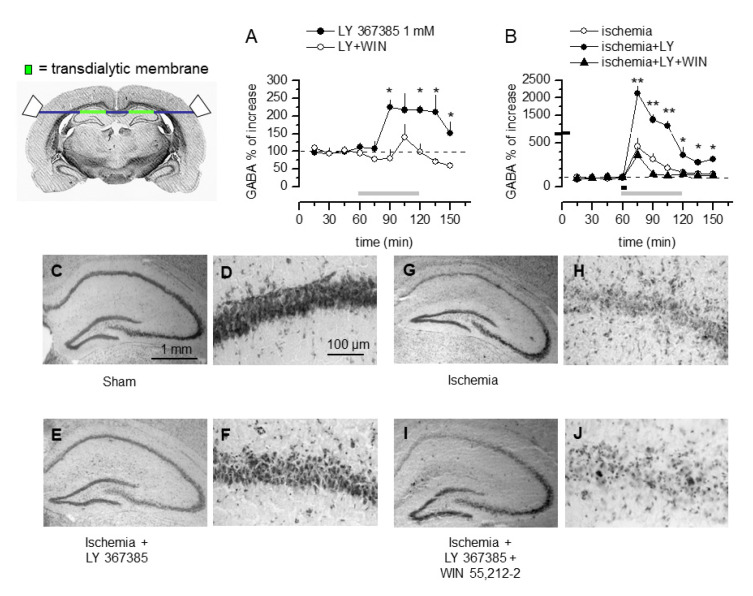
The hippocampal output of GABA and the neuroprotective effects produced by LY 367385 are prevented by WIN 55,212-2 in gerbils subjected to global ischemia. Top left: The position of the transerve microdialysis probe and of the transdialytic membranes is shown. (**A**,**B**) Time course of the changes in GABA concentrations induced by transmembrane perfusion of LY367385 (1 mM) and LY367385 + WIN 55,212-2 (50 µM) in hippocampal dialysates (gray lines depict time of application) of freely moving gerbils under control conditions (**A**) and after 5 min bilateral carotid occlusion (**B**, black line). GABA was measured in fractions collected every 15 min beginning at least 24 h after implantation of the dialysis tube and 2 h after equilibration washout. LY 367385 increases the basal output of GABA under both conditions and WIN 55,212-2 prevents this increase. * *p* < 0.05 and ** *p* < 0.01 vs. mean basal output, (ANOVA + Tukey’s w test). (**C**) Representative coronal sections from control and ischemic gerbils stained with toluidine blue. (**C**,**E**,**G**,**I**): Low magnification of hippocampal area. (**D**,**F**,**H**,**J**): High magnification of CA1 pyramidal cells. (**C**,**D**): Sham-operated controls. (**G**,**H**): Seven days following transient global ischemia, there is massive pyramidal cell degeneration in the CA1 area. (**E**,**F**): Ischemic gerbils transdialytically perfused treated with 1 mM LY 367385 display a dramatic reduction in CA1 damage. (**I**,**J**): Additional perfusion with 30 µM WIN 55,212-2 results in a reduction of the neuroprotective effect of LY367835.

**Figure 7 cells-11-03015-f007:**
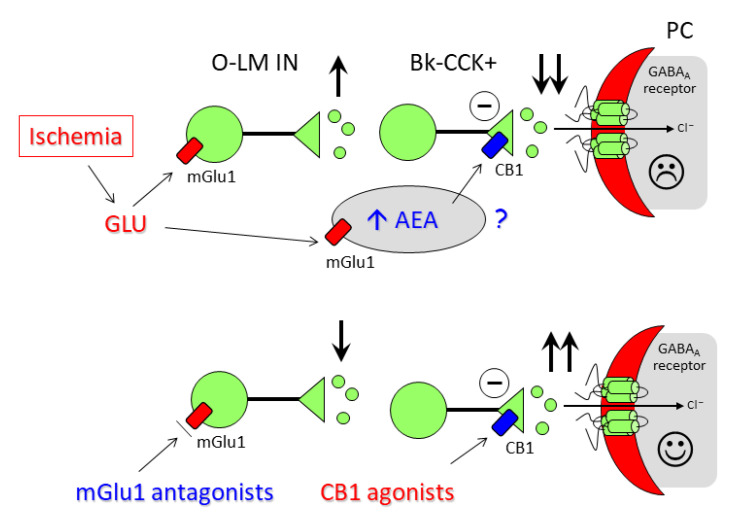
Polysynaptic GABAergic disinhibition model. mGlu1 are postsynaptic and CB1 receptors presynaptic in different interneuron populations connected in series. During ischemia, glutamate activates mGlu1 receptors in stratum oriens-stratum lacunosum moleculare GABAergic interneurons (O-LM IN) increasing their firing and leading to inhibition of CCK-positive basket cell (Bk-CCK+) that innervate the dendrites or the perisomatic region of CA1 pyramidal cells (PC), which results in a reduction in the release of GABA, disinhibition of excitatory inputs from the Schaffer collateral pathways and degeneration of CA1 pyramidal cells. mGlu1 antagonists will counteract this mechanism and, therefore, increase the net output of GABA upon pyramidal cells and provide neuroprotection, CB1 receptor agonists will reduce the output from basket cell terminals and prevent this effect.

## Data Availability

Our own data presented in this study are available on request from the corresponding author.

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
