# Peer review of "The Neuroprotective Effects of mGlu1 Receptor Antagonists Are Mediated by an Enhancement of GABAergic Synaptic Transmission via a Presynaptic CB1 Receptor Mechanism"

_cells, 2022, doi:10.3390/cells11193015_

Round 1

Reviewer 1 Report

The study entitled “The neuroprotective effects of mGlu1 receptors antagonists are mediated by an enhancement of GABAergic synaptic transmission via presynaptic CB1 receptor mechanism” from Dr. Pellegrini-Giampietro’s group addresses the question about actions of mGluR1 interacting with CB1 in the hippocampus in a model of ischemia. They report data to show that an mGlu1, but not mGlu5, receptor antagonist is able to increase synaptic inhibition (spontaneous inhibitory post-synaptic currents) in CA1 pyramidal cells. This effect is reduced by the activation of CB1 receptors. They show further that mGlu1 receptors antagonists can protect CA1 cells from ischemic damage, and this is reduced in presence of CB1 receptor activation, suggesting downstream activation of CB1 receptors.

Cross-talk between mGlu receptors and cannabinoid receptors has been shown in the CNS before, involving the production of lipid messengers. This study addresses circuits and mechanisms in a model of ischemia. Studies about functional interaction between receptors could have a relevance for the treatment of many diseases. The present study addresses an important topic and has the potential to move the field forward, but a number of concerns need to be addressed before it is suitable for publication.

One concern is that it is not entirely clear what experimental evidence is provided to support the site of action of mGluR1 in the circuitry proposed in Fig. 7 (“mGlu1 are postsynaptic and CB1 receptors presynaptic in different interneuron populations connected in series”). Spontaneous IPSC analysis in pyramidal cells (not interneurons) shows effects of mGlu1 antagonist (Fig. 1), and modulation by CB1 agonist (Fig. 2), on both frequency and amplitude distribution, which would indicate a pre- and post-synaptic site of action on pyramidal cells and is not captured in Fig. 7. They mention data on lack of effect of the mGlu1 antagonist on stratum oriens alveus interneurons (“did not show any changes in both frequency and amplitude of sIPSCs…and no depolarizing effect” line 296-298). These data should be shown, but it is not clear how they support the circuitry shown in Fig. 7.

Effects of mGlu1 and CB1 compounds on the kinetics of IPSCs, or lack thereof, should be shown.

The data for the experiments with WIN 55,212-2 should be shown in comparison with the effects of CP55,490 (Fig. 2). Likewise, effects of both compounds on frequency and amplitude of sIPSCs should be shown.

The electrophysiology is done in slices from normal animals but not in an ischemia model. This information is missing. It is not clear how mGlu1 would be activated endogenously under normal conditions as suggested by the effects of an mGlu1 antagonist in the electrophysiology experiments. Is there an increase in mGlu1 activation in the ischemic state? Endogenous mGlu1 activation under normal conditions apparently inhibits pyramidal cells (otherwise, the antagonist would not have any effect). Why would synaptic inhibition of pyramidal cells observed under normal conditions (Fig. 1 and 2) not be as “bad” as synaptic inhibition in the ischemic condition?  

The electrophysiology data do not really show an “interplay” (line 85) or “cross-talk” (line 21) between mGlu1 and CB1 and effects mediated “via presynaptic CB1” (title). What they show is opposing functions of mGlu1 and CB1 (effects of mGlu1 blockade can be inhibited/counteracted by CB1 activation). The HPLC and mass spec data support the involvement of endocannabinoids in mGlu1 effects. So, this is not a major concern, but it would be important to have electrophysiology data to support this.

A figure or image showing the implantation and sites of the dialysis probes would be useful.

Author Response

1st Query:

“One concern is that it is not entirely clear what experimental evidence is provided to support the site of action of mGluR1 in the circuitry proposed in Fig. 7 (“mGlu1 are postsynaptic and CB1 receptors presynaptic in different interneuron populations connected in series”). Spontaneous IPSC analysis in pyramidal cells (not interneurons) shows effects of mGlu1 antagonist (Fig. 1), and modulation by CB1 agonist (Fig. 2), on both frequency and amplitude distribution, which would indicate a pre- and post-synaptic site of action on pyramidal cells and is not captured in Fig. 7. They mention data on lack of effect of the mGlu1 antagonist on stratum oriens alveus interneurons (“did not show any changes in both frequency and amplitude of sIPSCs…and no depolarizing effect” line 296-298). These data should be shown, but it is not clear how they support the circuitry shown in Fig. 7.”

Response:

Our model shown in Fig. 7 is based on a number of anatomical and functional observations derived from our experiments and from studies of other laboratories. We are aware that a postsynaptic contribution of GABAA receptors to these mechanisms cannot be ruled out, but our “polysynaptic” hypothesis appears to be supported by most of the data that we are aware of. These observations and caveats are now listed in detail in a new paragraph in the Discussion (lines 605-620).

As for the experiments in interneurons, we would like to point out that many different types of interneurons are present in the stratum oriens-alveus in CA1. The interneurons that we patched in our control experiments may not have been somatostatin-positive O-LM interneurons expressing mGlu1 receptors. In case they were, the localization of mGlu1 and CB1 receptors in these cells is different from that of pyramidal cells, and it is possible that other types of experimental approaches are needed to detect changes in sIPSCs that are modulated by mGlu1 receptors. Because of these reasons, and because it would require an entirely novel figure for data that are negative, we would prefer to show the results in a descriptive format as they are now.

2nd Query:

“Effects of mGlu1 and CB1 compounds on the kinetics of IPSCs, or lack thereof, should be shown.”

Response:

As requested, we have inserted an inset in the new Fig.1A showing that the kinetics of IPSCs following the application of LY 367385 are not changed. The new data are presented in the Results section (lines 283-287) and discussed in lines 528-533.

3rd Query:

“The data for the experiments with WIN 55,212-2 should be shown in comparison with the effects of CP55,490 (Fig. 2). Likewise, effects of both compounds on frequency and amplitude of sIPSCs should be shown.”

Response:

As requested, we are now showing the experiments with WIN 55,212-2 in comparison with the effects of CP55,490 on frequency and amplitude of sIPSCs in the new Fig. 2D.

4th Query:

“The electrophysiology is done in slices from normal animals but not in an ischemia model. This information is missing. It is not clear how mGlu1 would be activated endogenously under normal conditions as suggested by the effects of an mGlu1 antagonist in the electrophysiology experiments. Is there an increase in mGlu1 activation in the ischemic state? Endogenous mGlu1 activation under normal conditions apparently inhibits pyramidal cells (otherwise, the antagonist would not have any effect). Why would synaptic inhibition of pyramidal cells observed under normal conditions (Fig. 1 and 2) not be as “bad” as synaptic inhibition in the ischemic condition?”

Response:

We have added in the Discussion a sentence clearly stating that electrophysiological experiments were performed on slices from non-ischemic rats and that a tonic activation of mGlu1 receptors exists (lines 526-528). We believe that this tonic stimulation is not harmful, but it leads to toxic mechanisms during ischemia, when stimulation and firing of mGlu1-containing neurons is enhanced by the dramatic increase in release of glutamate. This concept is now discussed on lines 621-623.

5th Query:

“The electrophysiology data do not really show an “interplay” (line 85) or “cross-talk” (line 21) between mGlu1 and CB1 and effects mediated “via presynaptic CB1” (title). What they show is opposing functions of mGlu1 and CB1 (effects of mGlu1 blockade can be inhibited/counteracted by CB1 activation). The HPLC and mass spec data support the involvement of endocannabinoids in mGlu1 effects. So, this is not a major concern, but it would be important to have electrophysiology data to support this.

Response:

The electrophysiology data represent only the initial part of our study and were designed to understand the basal circuitry containing mGlu1 or CB1 receptors converging on CA1 pyramidal cells. Our concept of “interplay” or “cross-talk” between mGlu receptors and endocannabinoids following ischemia was based mostly on our neuroprotection, GABA release, immunocytochemistry and endocannabinoid quantitation studies. We agree that it would be important to have electrophysiological data on post-OGD slices to support this, but it is quite difficult to perform whole cell patch-clamp experiments in cells that are damaged following simulated ischemia. Typically, field potential but not single cell recordings are carried out in hypoxic slices.

6th Query:

“A figure or image showing the implantation and sites of the dialysis probes would be useful.”

Response:

As requested, we have included an image of the microdialysis probe position in Fig. 6.

Reviewer 2 Report

The manuscript entitled “The neuroprotective effects of mGlu1 receptor antagonists are mediated by an enhancement of GABAergic synaptic transmission via a presynaptic CB1 receptor mechanism”, is a well-written article that highlight some important considerations regarding the neuroprotective properties of mGlu1 antagonists and a functional interaction between mGlu1 receptors and the endocannabinoid system in models of cerebral ischemia.

Authors have performed a series of multiple in vitro and in vivo experiments, including electrophysiological, morphological, and release studies in hippocampal tissue exposed to oxygen and glucose deprivation, as well as in ischemic conditions. Presented analysis of results revealed that the neuroprotective effects of mGlu1 receptor antagonists are mediated by an enhancement of GABAergic neurotransmission that appears to be regulated by the endocannabinoid system - presumably activation of presynaptic CB1 receptors.  

Taken together, these results provide valuable insight into the involvement of mGlu1 receptors and the endocannabinoid system in the mechanism of the post-ischemic neuronal death.

I would like to recommend a minor revision in the text.

1.     Figure 5. On the graph 5(B) the basal level of 2.AG according to the scale on Y axis is around 330 ng/mg, whereas in the results (section 3.5) authors stated that under the basal conditions, the level of 2-AG was 294+/-25 ng/mg (text line 424). Could you clarify this, please.

2.     Figure 2. Could you please refine the position of the capital letter (A,B,C,D,E) that depicts particular graphs on the figure, to match them better with corresponding graphs.

3.     Please add in the figures legend 2(C), 3(C and D) what is the meaning of numbers in parenthesis that are presented within or above the bars in particular graphs (similar to the description provided in the figure legend 1(D,E): “The number of cells tested is in parenthesis”).

Author Response

1st Query:

“Figure 5. On the graph 5(B) the basal level of 2.AG according to the scale on Y axis is around 330 ng/mg, whereas in the results (section 3.5) authors stated that under the basal conditions, the level of 2-AG was 294+/-25 ng/mg (text line 424). Could you clarify this, please.”

Response:

We are sorry for the mistake in the text. We have inserted the correct concentration of 2-AG under basal conditions (341±69 ng/mg of protein) in the Results section 3.5.

2nd Query:

“Figure 2. Could you please refine the position of the capital letter (A, B, C, D, E) that depicts particular graphs on the figure, to match them better with corresponding graphs.”

Response:

We are sorry again for the mistake. We have corrected the positions of the letters in Fig. 2.

3rd Query:

Please add in the figures legend 2(C), 3(C and D) what is the meaning of numbers in parenthesis that are presented within or above the bars in particular graphs (similar to the description provided in the figure legend 1 (D, E): “The number of cells tested is in parenthesis”).

Response:

As requested, we have introduced the meaning of the numbers in parenthesis for Figs. 2 and 3.

Reviewer 3 Report

The study by Landucci et al cleary shows the neuroprotective effect of mGlu1 receptor antagonists in in vitro and in vivo models of ischemia. This neuroprotective effect is mediated by an enhancement of GABAergic neurotransmission that seems to be regulated by endocannabinoid-mediated activation of CB1 receptors. In vitro and in vivo analysis shows that mGlu1 receptor antagonists enhance sIPSCs and protect against post-ischemic injury in CA1 pyramidal cells and in hippocampus through the reduction of CB1 receptor activation. Authors also analysed the contents of anandamide and 2-arachidonoylglycerol in hippocampal slices exposed to oxygen and glucose deprivation and in ischemic gerbils, finding an increase in anandamide levels which is reversed by the use of the mGlu1 receptor antagonist. The manuscript is well written and structured; in vivo and in vitro experiments clearly show the cooperative interaction between the glutamate and endocannabinoid system during ischemia.

-Knowing that group I mGlu receptors comprise mGlu1 and mGlu5 receptor and that they have a pleiotropic effect, it would be interesting to clarify if mGlur5 receptor also participates in the neuroprotective/neurotoxic action in ischemic models (e.g. MTEP could be used in OGD toxicity or other)

- minor

Please check the CP 55940 compound it is written in different ways on page 9

Typos:

Line 52     2-aracidonoylglicerol

Line 102    double space

Figure 1 in the caption (C) is reported 300 mM I think it is 100 mM

Author Response

1st Query:

“Knowing that group I mGlu receptors comprise mGlu1 and mGlu5 receptor and that they have a pleiotropic effect, it would be interesting to clarify if mGlur5 receptor also participates in the neuroprotective/neurotoxic action in ischemic models (e.g. MTEP could be used in OGD toxicity or other).”

Response:

The role of mGlu5 receptors in models of ischemia is described in the first paragraph of the Introduction: the use of antagonists such as MPEP is unable to attenuate OGD-induced injury whereas the use of mGlu5 PAMs leads to neuroprotection. We have added a brief statement on the effects of MPEP. We have not tested the effects of MTEP.

2nd Query:

Minor points:

Please check the CP 55940 compound it is written in different ways on page 9

Typos: Line 52: 2-aracidonoylglicerol, Line 102: double space, Figure 1 in the caption (C) is reported 300 mM I think it is 100 mM

Response:

We have corrected the indicated mistakes, but in the caption of Fig. 1C it is indeed 300 µM LY367385.

Round 2

Reviewer 1 Report

I am satisfied by the way the authors have addressed my concerns